# The Role of Psychological Interventions in Enhancing Quality of Life for Patients with Cystic Fibrosis—A Systematic Review

**DOI:** 10.3390/healthcare13091084

**Published:** 2025-05-07

**Authors:** Lavinia Hogea, Brenda Bernad, Iuliana Costea, Codrina Mihaela Levai, Amalia Marinca, Ion Papava, Teodora Anghel

**Affiliations:** 1Neuroscience Department, “Victor Babes” University of Medicine and Pharmacy, 300041 Timisoara, Romania; hogea.lavinia@umft.ro (L.H.); papava.ion@umft.ro (I.P.); anghel.teodora@umft.ro (T.A.); 2Neuropsychology and Behavioral Medicine Center, “Victor Babes” University of Medicine and Pharmacy, 300041 Timisoara, Romania; 3Psychology Department, West University of Timisoara, 300223 Timisoara, Romania; 4Discipline of Medical Communications, Department 2—Microscopic Morphology, “Victor Babes” University of Medicine and Pharmacy, 300041 Timisoara, Romania; codrinalevai@umft.ro; 5Center for Studies and Research in Psychology, Faculty of Psychology, “Tibiscus” University, Lascăr Catargiu 4-6, 300559 Timisoara, Romania; amalia_marinca@yahoo.com

**Keywords:** cystic fibrosis, psychological interventions, quality of life

## Abstract

**Background/Objectives:** Cystic fibrosis (CF) is a chronic genetic disease that impacts both physical and psychological health, increasing vulnerability to anxiety, depression, and reduced quality of life (QoL). Psychological interventions, particularly cognitive behavioral therapy (CBT), have demonstrated promising results in enhancing emotional resilience, treatment adherence, and QoL. This systematic review aims to evaluate the role and effectiveness of psychological interventions in improving the QoL among individuals with CF. **Methods:** A comprehensive literature search was conducted across the PubMed, Scopus, and PsycINFO databases for studies published between 2014 and 2024, in line with PRISMA guidelines and a registered PROSPERO protocol. Out of 162 initially identified articles, six clinical studies met the inclusion criteria. Intervention included cognitive behavioral therapy-based interventions, employing several digital or telehealth formats such as fibrosis-specific cognitive behavioral therapy (CF-CBT) and the coping and learning to manage stress (CALM) program, often delivered via telehealth. **Results:** Most interventions demonstrated significant reductions in depression, anxiety, and perceived stress, alongside improvements in coping self-efficacy and vitality. Cohen’s d-effect sizes ranged from moderate to large for core psychological outcomes. QoL measures, particularly vitality and emotional functioning, were significantly enhanced in most studies. **Conclusions:** Psychological interventions, particularly CBT and ACT, significantly improve mental health and QoL in individuals with CF, supporting their integration into routine care.

## 1. Introduction

Cystic fibrosis (CF) is an autosomal recessive genetic disorder caused by mutations in the cystic fibrosis transmembrane conductance regulator (CFTR) gene, located on chromosome 7, which regulates the transport of chloride and bicarbonate ions across epithelial membranes. When both gene copies are mutated, it leads to multi-organ issues, mainly in the lungs and digestive system. CF affects about 1 in 2500 live births, especially among Caucasians [1,2].

Effective management of CF requires a multidisciplinary approach involving pulmonologists, nutritionists, physical therapists, mental health professionals, and social workers. This team collaborates to create individualized treatment plans that address airway clearance, infection control, nutritional support, and psychosocial well-being, as each aspect of care is interconnected [3]. Recent advancements, such as CFTR modulators, have transformed treatment by targeting defective CFTR proteins, improving respiratory function and quality of life (QoL). These therapies emphasize precision medicine, tailoring treatments to specific genetic mutations and requiring close monitoring for side effects [4]. This integrated approach ensures comprehensive care for the diverse presentations of CF.

Chronic stress from CF takes a heavy psychological, emotional, and physical toll on patients and families. Daily symptoms and time-consuming treatments (2–4 h/day) contribute to ongoing distress, even as life expectancy improves, now around 40 years, and potentially over 50 with newer therapies [5]. Patients with CF face elevated rates of anxiety (26.22%) and depression (14.13%), driven by chronic symptoms, financial burdens, and social isolation. QoL is significantly reduced in CF patients, with lower scores in both physical and psychosocial health domains [6]. Holistic palliative care, like the CF-CARES model, supports emotional, psychological, and spiritual well-being, boosting QoL and treatment adherence. Family, peer, and community support strengthen resilience, especially in young patients [7].

Psychological interventions, particularly cognitive behavioral therapy (CBT), are essential in managing CF’s intertwined medical and psychological challenges. CF patients often face significant psychological stressors, with anxiety and depression rates ranging from 5% to 22%, alongside the physical demands of rigorous treatment regimens [7].

Guidelines from the Cystic Fibrosis Foundation and the European Cystic Fibrosis Society emphasize regular mental health screenings during routine care, as co-occurring mental health issues can reduce medication adherence, worsen lung function, and diminish health-related quality of life (HRQoL) [8]. CBT effectively reduces anxiety and depression, leading to better treatment adherence and health outcomes [9]. Adolescents and young adults with CF often struggle with treatment adherence due to emotional distress. Integrated CBT programs, especially when combined with psychoeducation, improve mental well-being, reduce isolation and anxiety, and promote healthier coping and compliance [7].

Acceptance and commitment therapy (ACT) is a practical approach for managing the emotional challenges of chronic illnesses like CF. By promoting psychological flexibility and acceptance of complex thoughts and feelings, ACT helps patients cope with symptoms, treatment demands, and health uncertainties, reducing anxiety and depression [10]. ACT also boosts self-efficacy and motivation, empowering patients to manage their treatment regimens and navigate healthcare systems more effectively. Mindfulness practices within ACT further help patients cope with anticipatory anxiety about disease progression and treatment outcomes [10]. Research shows that ACT empowers patients to manage treatment better and navigate the healthcare system. Its mindfulness components help reduce anticipatory anxiety around disease progression and outcomes [10]. In the long term, ACT enhances QoL and longevity by alleviating treatment-related stress and promoting better health management [10].

Mindfulness and relaxation techniques (MRT) are increasingly valued as psychological interventions for CF patients, offering support for mental health, well-being, and coping strategies [11,12,13]. These practices are fundamental, given the psychological distress linked to chronic conditions like CF, which can hinder treatment adherence and QoL. Research shows that mindfulness improves emotional regulation and reduces anxiety and depression, which are common among CF patients. Tailoring mindfulness interventions to address CF-specific challenges further enhances their effectiveness, improving health outcomes [13].

Social support also plays a vital role in CF management. Studies have shown that robust social networks are associated with improved health outcomes, including reduced symptom severity and treatment burden. Support from family, friends, and healthcare providers eases psychological strain and boosts overall well-being [14].

Existing studies highlight the mental health challenges faced by CF, including elevated levels of anxiety and depression, which are associated with poorer clinical outcomes such as reduced lung function and diminished QoL [15,16,17,18,19]. Recognizing this, key CF organizations now recommend routine psychological screening as a standard component of care, particularly for patients aged 12 years and older, to facilitate early identification and intervention for psychological distress [16,20,21,22]. These screenings are increasingly integrated into multidisciplinary care teams, improving patient engagement and overall health-related QoL [7]. Despite the growing recognition of mental health care as a critical component of CF management, the literature remains fragmented, with significant variations in intervention types and outcomes complicating the establishment of best practices. This underscores the need for a systematic review following PRISMA guidelines to synthesize existing evidence, evaluate the effectiveness of psychological interventions, and identify gaps in accessible mental health resources, ultimately guiding future research and improving holistic CF care.

This study aims to conduct a systematic literature review to evaluate the effectiveness of psychological interventions in enhancing the QoL for patients with CF. It will analyze existing studies with the primary objective of identifying the main types of psychological interventions, assessing their impact on mental health and QoL in CF patients, highlighting research gaps, and proposing future directions for more effective interventions.

## 2. Materials and Methods

This paper presents a systematic review focused on investigating the effectiveness of psychological interventions in improving the QoL of patients with CF. The objective is to identify and analyze studies conducted over the past decade, following the guidelines outlined in the PRISMA (Preferred Reporting Items for Systematic Reviews and Meta-Analyses) protocol [23,24]. A protocol for this article has been registered in the PROSPERO database under the ID number CRD420251029524 (https://www.crd.york.ac.uk/PROSPERO/view/CRD420251029524, accessed on 10 April 2025). To ensure accuracy and clarity in writing, ChatGPT (version GPT-4.5) was used for content refinement and Grammarly (version 14.1227.1) for grammatical and stylistic correction.

### 2.1. Eligibility Criteria

This systematic review encompasses studies on individuals with CF, regardless of age, gender, or disease severity, and those with comorbid psychological conditions, provided the intervention is relevant. Eligible studies examine psychological interventions to improve QoL, including CBT, mindfulness, ACT, relaxation techniques, stress management, and online support programs. Studies must include control groups that receive standard care, a placebo, or alternative psychological treatments.

The primary outcome is improvement in QoL, assessed through standardized measures, including CF-specific QoL questionnaires, SF-36, and WHOQOL. Secondary outcomes include psychological well-being, depression, anxiety, treatment adherence, and coping mechanisms. Only randomized controlled trials (RCTs), longitudinal, and observational studies published in peer-reviewed journals between 2014 and 2024 are included. Qualitative studies are included if they offer valuable perspectives on the emotional and psychological experiences of individuals with CF, such as how they cope with chronic illness, manage stress and anxiety, perceive their QoL, and navigate mental health challenges within the context of their daily treatment routines.

Exclusion criteria apply to studies focusing solely on caregivers, pharmacological treatments, or non-CF-specific mental health interventions. Studies lacking measurable QoL or psychological outcomes, case reports, conference abstracts, editorials, and those with methodological weaknesses (e.g., no control group or high bias risk) will not be considered. Non-peer-reviewed sources, gray literature, and unpublished dissertations are also excluded.

### 2.2. Selection Process

Two independent reviewers screened all retrieved records based on title and abstract, assessing their relevance to the research question. Studies that met the preliminary inclusion criteria were then subjected to a full-text review, again conducted independently by both reviewers. Any discrepancies in study selection were resolved through discussion, and if a consensus could not be reached, a third reviewer was consulted to make the final decision. Automation tools, such as Rayyan, were used to facilitate the initial screening and deduplication process, helping to streamline the selection workflow and minimize bias.

### 2.3. Data Collection Process

Extracted information from the selected articles, including study design, sample characteristics, type of psychological intervention, outcome measures related to QoL and psychological well-being, and statistical results. Two independent reviewers systematically reviewed and coded the data to ensure accuracy and reliability, resolving discrepancies through discussion. Standardized extraction forms were used to capture relevant details, and automation tools, such as Rayyan, facilitated data organization and screening. Quantitative and qualitative data were synthesized to assess the impact of psychological interventions on CF patients, ensuring a comprehensive evaluation of study outcomes

### 2.4. Risk of Bias

While most studies employed rigorous methodologies, some limitations include small sample sizes, a lack of long-term follow-up, potential self-report biases, and variability in implementing interventions. Additionally, differences in study designs and outcome measures may introduce heterogeneity, which can affect the overall reliability of the findings.

### 2.5. Effect Measure and Synthesis Methods

The effect measures for this systematic review included Cohen’s d to assess the impact of psychological interventions on QoL, depression, anxiety, and adherence outcomes in CF patients. Data synthesis was conducted using a narrative approach for qualitative findings.

## 3. Results

The search strategy systematically identified relevant studies across PubMed, Scopus, and PsycINFO using predefined keywords and Boolean operators to capture psychological interventions for patients with CF: (“cystic fibrosis”) AND (“psychological interventions” OR “cognitive behavioral therapy” OR “CBT” OR “ACT” OR “mindfulness” OR “relaxation techniques” OR “stress management”) AND (“quality of life” OR “QoL”). Only clinical, peer-reviewed studies published in English between 2014 and 2024 were included. The PubMed search retrieved 143 articles, of which 134 were excluded due to relevance, study design, or methodological quality, leaving nine eligible studies. Scopus identified 17 articles, all of which were excluded for irrelevance. In contrast, PsycINFO retrieved two studies, both of which were excluded due to methodological limitations. The PRISMA flowchart illustrates this process, showing 161 records identified, with one duplicate removed and 136 excluded during screening, because they were studies that consisted primarily of systematic reviews, meta-analyses, papers addressing cystic fibrosis without incorporating psychological interventions, or studies involving psychological interventions applied to other chronic conditions rather than cystic fibrosis. This resulted in six final studies in the systematic review, ensuring a rigorous and transparent selection process (Figure 1).

The reviewed papers encompassed a diverse range of participants, with 362 individuals across six studies (Table 1). Gender-specific information was limited; however, among the studies reporting gender, the female participants (n = 7) slightly outnumbered the males (n = 3) in the pilot trial by Verkleij et al. (2023) [25]. Sample sizes varied significantly, with the smallest pilot study enrolling 10 participants (Verkleij et al., 2023) [25] and the largest randomized controlled trials (RCTs) enrolling up to 132 participants (Bathgate et al., 2024) [26].

The studies reviewed primarily included randomized controlled trials (RCTs), comparative effectiveness trials, and pilot studies. Four studies employed RCT designs, including those by C. V. O’Hayer et al. (2024) [10] and Bathgate et al. (2024) [26]. In contrast, the remaining studies by Lorenc et al. (2015) [26], Friedman et al. (2022) [27], Verkleij et al. (2021) [27], and Verkleij et al. (2023) [25] were categorized as comparative effectiveness and pilot trials, reflecting initial exploratory investigations.

Intervention modalities were predominantly telehealth-based, utilizing CBT approaches. Three studies implemented CF-CBT delivered via telehealth or digitally (Friedman et al., 2022; Verkleij et al., 2021, 2023) [25,27,28]. One study tested a CALM program via telehealth sessions (Bathgate et al., 2024) [26], and one, ACT via telehealth (O’Hayer et al., 2024) [10]. Additionally, one study explored Tai Chi as an integrative approach (Lorenc et al., 2015) [29], providing an alternative, non-traditional therapeutic option.

The duration of interventions and follow-up assessments ranged from unspecified in the CF-CBT pilot trials to nine months in the Tai Chi comparative trial (Lorenc et al., 2015) [29]. The RCTs by O’Hayer et al. (2024) [10] and Bathgate et al. (2024) [26] featured a consistent 3-month follow-up, emphasizing short-term to medium-term outcomes.

The reported outcomes primarily included reductions in depression, anxiety, and perceived stress, along with improvements in coping self-efficacy, vitality, and QoL. Measures related to QoL frequently utilized standardized instruments such as the Cystic Fibrosis Questionnaire–Revised (CFQ-R) and various psychological scales (e.g., GAD-7, PHQ-9).

Significant findings were generally positive across the interventions. The telehealth interventions (CALM and ACT) demonstrated substantial improvements in psychological outcomes, notably significant reductions in depression and anxiety symptoms, both immediately post-treatment and at follow-up. For instance, Bathgate et al. (2024) reported Cohen’s d of 0.85 for depression post-intervention and 0.70 at follow-up, indicating strong clinical relevance [26]. Similarly, Verkleij et al. (2023) noted robust improvements, with Cohen’s d values indicating significant effects on depression (PHQ-9 = −1.23) and anxiety (GAD-7 = −1.09) [25].

The statistical analysis of Cohen’s d effect sizes across the reviewed studies highlights the substantial effects of psychological interventions on depression, anxiety, stress, and QoL in individuals with CF (Figure 2). Depression showed strong post-treatment effects, with an average Cohen’s d of 0.94, reflecting a consistent and robust improvement ranging from moderate (0.83) to large (1.23). Follow-up assessments for depression indicated a stable and moderate effect (d = 0.7), demonstrating sustained benefits. Similarly, anxiety showed substantial immediate post-treatment improvements, averaging Cohen’s d = 0.80, and stable follow-up results (d = 0.66), suggesting durable reduction in anxiety. Additional measures from single studies, such as stress reduction (d = 0.73), QoL vitality (d = 1.11), and relaxation skills (d = 0.93), further supported the effectiveness of the interventions, although these outcomes lacked comparative data across multiple studies. The analysis highlights the consistently positive impact and clinical relevance of these interventions on psychological outcomes for CF patients.

**Table 1 healthcare-13-01084-t001:** Randomized control trial studies of psychotherapy among cancer patients.

Study (Year and Country)	Number of Participants	Type of Study	Intervention	Duration	Outcome	Measures Related to QoL	Significant Results	Cohen’s d
Verkleij, 2021, Netherlands [21]	10 (not specified by gender)	Pilot study	Therapist-guided eHealth CF-CBT	Intervention spanned eight sessions (exact weeks not specified, but typically 8 weeks)	Assessed feasibility, usability, acceptability, and preliminary efficacy in reducing depression, anxiety, perceived stress, and improving QoL	Coping self-efficacy, stress, vitality scores	90% of participants improved on GAD-7 (anxiety) 90% improved on PHQ-9 (depression), 80% improved on PSS (perceived stress) 70% showed improvement in the CFQ-R health perception	Not specified
Lorenc, 2015, United Kingdom [29]	72	Comparative effectiveness trial	Eight Tai Chi sessions	9 months	Impact on QoL, mindfulness, and sleep	Health-related QoL	Not yet reported	Not specified
Friedman, 2022, Netherlands [28]	14 (13 completed, one discontinued)	Pilot study	Eight sessions of CF-CBT delivered in person or via telehealth	Not specified	Improvements in depressive symptoms, stress, coping	Cystic Fibrosis Questionnaire–Revised (CFQ-R), perceived stress, coping confidence	Large effect size for depressive symptoms (−0.83), CFQ-R vitality (1.11), relaxation skills (0.93)	Not specified
Verkleij, 2023, Netherlands [25]	10 (7 female, three male)	Pilot trial	Eight sessions of therapist-guided digital CF-CBT	Not specified	Reduction in depression, anxiety, and stress, improved QoL	CFQ-R health perceptions, vitality	90% improvement in GAD-7 (50% ≥ MID of 4 points), 90% improvement in PHQ-9 (40% ≥ MID of 5 points)	PHQ-9 (−1.23), GAD-7 (−1.09), PSS (−0.73)
O’Hayer, 2024, USA [10]	124 (93 female, 31 male)	RCT	Six sessions of ACT delivered via telehealth	3-month follow-up	Reduction in depression, anxiety, cognitive fusion	Psychological flexibility, acceptance, QoL	ACT reduced depression (BDI-II), anxiety (BAI), and cognitive fusion	0.59
Bathgate, 2024, USA [26]	132 (66 immediate treatment, 66 waitlist)	RCT	Six sessions of telehealth intervention + booster	3-month follow-up	Reduction in depression, anxiety, and perceived stress; improvement in coping self-efficacy and vitality	Coping self-efficacy, stress, vitality scores	The immediate group reported significantly lower depression, anxiety, and stress post-intervention and at 1-month follow-up.	Depression (0.85 post-intervention, 0.70 at follow-up); anxiety (0.65 post-intervention, 0.66 at follow-up)

The reviewed studies present degrees of risk of bias due to missing results, primarily arising from reporting biases. The extensive randomized controlled trials, such as those conducted by O’Hayer et al. (2024) [10] and Bathgate [26], explicitly reported comprehensive follow-up data with detailed statistical outcomes, indicating a relatively lower risk of bias from missing results. However, pilot studies, including those by Friedman [28] and Verkleij [25], demonstrated incomplete reporting of long-term follow-up effects and lacked detailed statistical metrics for some outcomes, which potentially increased the risk of bias due to selective or incomplete reporting. Additionally, the comparative effectiveness trial by Lorenc [29] has not yet reported significant results, introducing uncertainty and an elevated risk of bias due to missing outcome data.

Certainty in evidence for depression and anxiety outcomes is high due to consistent findings across multiple robust RCTs. However, certainty is moderate to low for stress and specific QoL outcomes, given the limited number of studies and smaller sample sizes. Further research is needed to strengthen confidence in these secondary outcomes.

## 4. Discussion

This systematic review analyzed six clinical studies to assess the effectiveness of psychological interventions in enhancing QoL among patients with CF. The studies included 362 participants and primarily utilized telehealth-delivered CBT, ACT, and CALM programs. These interventions consistently demonstrated significant improvements in depression, anxiety, and perceived stress, alongside notable gains in coping self-efficacy and vitality. CBT-based interventions, especially CF-specific versions delivered digitally, and ACT showed strong clinical relevance with moderate to large effect sizes.

The review found that the effect sizes for psychological outcomes were robust, with Cohen’s d ranging from 0.59 to 1.23 across measures like depression (e.g., PHQ-9), anxiety (e.g., GAD-7), stress (PSS), and QoL indicators like vitality. For instance, the CALM and CF-CBT programs achieved effect sizes of 0.85 and above for depression post-intervention, suggesting a strong treatment effect. Although pilot studies generally lacked long-term follow-up, larger RCTs confirmed the durability of treatment effects up to three months. The ACT intervention, in particular, enhanced psychological flexibility and emotional acceptance, adding a valuable dimension to CF care.

Psychological interventions for individuals with CF are essential due to the high prevalence of mental health issues associated with this chronic illness [8,30]. CF presents a significant psychological burden, with studies indicating that the incidence of depression and anxiety among patients and their families is notably high [31,32,33]. Mental health issues can adversely affect treatment adherence, QoL, and long-term health outcomes [17]. To mitigate these challenges, international guidelines have been developed to outline the integration of mental health screening and interventions within standard CF care practices [34].

The Cystic Fibrosis Foundation and the European Cystic Fibrosis Society recommend that all individuals with CF aged 12 and above undergo annual screenings using validated tools, such as the Patient Health Questionnaire (PHQ-9) and the Generalized Anxiety Disorder 7-item scale (GAD-7) [8,17]. These guidelines emphasize early identification and subsequent treatment of mental health issues, which can be addressed through psychological counseling and, in some cases, pharmacological interventions [17]. Implementing routine mental health screening has proven beneficial in tracking psychological well-being and facilitating timely intervention [34].

Studies show mixed adherence outcomes of ACT in CF, with objective data often revealing lower adherence than self-reports, complicating assessments of ACT’s effectiveness [7,35]. In contrast, CBT has a stronger evidence base for improving mental health and, in turn, adherence among CF patients, particularly by addressing depression and anxiety [11,36]. While both ACT and CBT aim to enhance psychological flexibility, ACT emphasizes mindfulness and acceptance, whereas CBT targets negative thought patterns to improve behavior [20]. Some research suggests combining both may better support adherence by addressing CF’s emotional and practical challenges [37,38].

Literature shows a substantial body of evidence supporting the efficacy of telehealth-delivered psychological interventions, particularly CBT, in improving mental health outcomes in individuals with chronic illnesses, including CF; recent findings in O’Hayer et al., 2024 study, also demonstrate that ACT is effective in this context, showing superiority over support psychotherapy in reducing psychological distress and enhancing PF in adults with CF [10]. Also, CBT has been linked to increased adherence rates to prescribed therapies, which is critical given the complex regimen that CF patients must follow. Adherence is directly correlated with better lung function and reduced hospitalizations due to exacerbation, underscoring the integral role of mental health support in the clinical management of CF [11,39].

Implementing telehealth-based psychological care for CF in low-resource settings faces significant challenges due to infrastructural gaps, socio-economic disparities, and cultural barriers. Limited internet access and insufficient technology or digital literacy impede effective telehealth delivery [40,41,42,43,44]. Building trust through virtual platforms is also tricky, especially when in-person rapport is valued, affecting patient engagement [45,46,47]. Financial constraints further hinder adoption, as facilities often lack the resources to invest in technology or training, and reimbursement for tele-psychological care remains inconsistent [48,49]. Additionally, socio-economic and cultural factors—including stigma and access to devices—further limit telehealth’s reach among vulnerable CF populations [42,45].

Quittner et al. (2016) emphasized the importance of integrating mental health services into CF care, highlighting that evidence-based interventions such as CBT significantly reduce the symptoms of anxiety and depression when tailored to the unique challenges of CF patients [16]. Similarly, Georgiopoulos et al. (2021) found that telehealth delivery enhances accessibility and adherence, offering a feasible and acceptable mode of treatment without compromising therapeutic impact [50]. The results are in concordance with these findings. The reviewed studies consistently reported clinically significant reductions in depression and anxiety following telehealth-based interventions, including CF-specific CBT [25,27], ACT [10], and the CALM program [26]. Notably, the effect sizes for depression and anxiety in the analysis ranged from moderate to large (Cohen’s d = 0.65 to 1.23), aligning with prior meta-analytic evidence that demonstrates the strong impact of CBT on emotional distress in chronic illness populations [51,52]. For instance, Hofmann et al. (2012) reported an average effect size of 0.88 for CBT in treating anxiety and 0.62 for depression, which closely mirrors the range observed in the synthesis [52].

Moreover, studies in this review confirm the importance of targeting stress and QoL in CF care, which is also echoed in earlier work by Riekert et al. (1998) and Smith et al. (2016), who demonstrated that psychological flexibility and coping self-efficacy are essential mediators of improved QoL [17,53]. The reported Cohen’s d values for QoL-related outcomes, such as vitality (d = 1.11) and relaxation skills (d = 0.93), reinforce the impact of behavioral interventions on broader well-being indicators. However, limited comparative data prevent definitive conclusions. This finding is partially concordant with those from other chronic illness contexts, where non-CBT interventions, such as mindfulness-based stress reduction (MBSR), have shown promising effects on vitality and life satisfaction [12].

Gender significantly influences psychological intervention outcomes in CF, with women often facing earlier disease complications, lower adherence, and shorter life expectancy due to both biological and sociocultural factors [54,55]. Despite reporting higher psychological distress, women may avoid mental health services due to stigma or fear of judgment, while men may resist showing vulnerability, affecting care engagement [56]. These patterns highlight the need for gender-sensitive approaches considering differing psychosocial experiences to improve intervention effectiveness [57].

However, findings also highlight areas of discrepancy or limited evidence compared to the broader literature. While some literature reports sustained long-term effects of CBT and ACT interventions of up to 6–12 months [7,10,58], the follow-up periods in most included CF studies were limited to 3 months, and longer-term outcomes remain underreported. This discrepancy suggests a need for future studies with extended follow-up to assess the durability of treatment effects.

Literature suggests that mind–body practices such as Tai Chi and mindfulness can significantly improve psychological well-being and reduce symptoms of stress, anxiety, and depression in individuals with chronic illnesses [59,60,61]. Building on this foundation, CF studies could explore the applicability and effectiveness of these interventions in improving QoL and mental health outcomes. The Tai Chi intervention [29] has yet to demonstrate significant results, which contrasts with the literature on mind–body practices, which typically reports modest but positive effects on mood and QoL [60].

This study presents several limitations. Most notably, the sample sizes in several included studies were relatively small, particularly in pilot trials, which may reduce the generalizability of findings and limit statistical power to detect subtle effects. Additionally, gender-specific data were often underreported, and long-term outcomes beyond three months were rarely assessed, restricting conclusions about sustained benefits. Despite these limitations, the findings have meaningful applicability in clinical practice. Telehealth-delivered psychological interventions, especially CBT and ACT-based approaches, show strong potential for integration into standard CF care protocols, particularly in settings where access to in-person therapy is limited. However, to standardize their implementation, formal guidelines and regulatory frameworks are needed to define therapeutic protocols, credentialing requirements for practitioners, and ethical standards for digital delivery.

Future research should prioritize large-scale, multicenter, randomized controlled trials with extended follow-up periods and diverse patient populations to strengthen the evidence base. Additionally, exploring the comparative effectiveness of different psychological modalities and evaluating their cost-effectiveness will be essential for informing policy and ensuring sustainable implementation in clinical settings.

## 5. Conclusions

This systematic review highlights the effectiveness of psychological interventions, CBT, and ACT, in enhancing mental health and QoL for individuals with CF. Among the six clinical studies included from 2014 to 2024, telehealth-delivered interventions consistently reduced symptoms of depression, anxiety, and stress. Significant improvements were also noted in coping self-efficacy, vitality, and emotional functioning—key components of QoL. Programs like CF-CBT and CALM demonstrated large effect sizes, underlining their strong clinical relevance. While evidence for integrative approaches such as Tai Chi and mindfulness is still emerging, the findings support incorporating psychological care into routine CF treatment. These interventions enhance psychological well-being and significantly contribute to overall QoL, underscoring their crucial role in comprehensive CF care.

## Figures and Tables

**Figure 1 healthcare-13-01084-f001:**
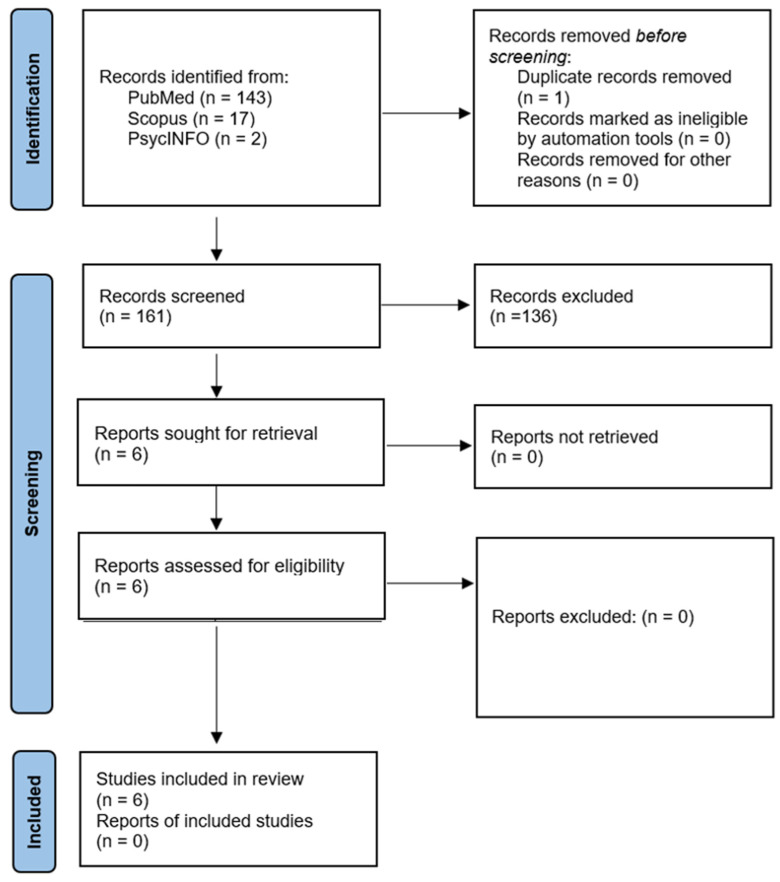
PRISMA flowchart of selected papers for the study.

**Figure 2 healthcare-13-01084-f002:**
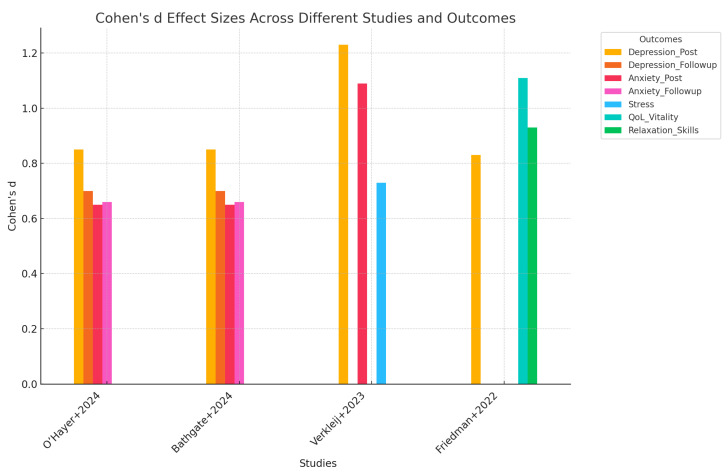
Cohen’s d effect sizes across different studies and outcomes [10,27,28,29].

## Data Availability

All data discussed in this review are available in the original publications cited in the reference list.

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
