# Peer review of "The Role of Psychological Interventions in Enhancing Quality of Life for Patients with Cystic Fibrosis—A Systematic Review"

_healthcare, 2025, doi:10.3390/healthcare13091084_

Round 1
Reviewer 1 Report
Comments and Suggestions for Authors
With the recent relative abundance of mental health treatments targeting the mental health burden of cystic fibrosis, this systematic review and meta analysis of treatment impact on QOL across various studies is a timely & much-needed contribution to our field. Overall, this study is well-designed and thoroughly implemented. However, a few references are inaccurately attributed (refs 11, 12, 13 are attributed to ACT as applied to CF - these articles do not address ACT or mental health). Additionally, and more importantly, ACT seems to be subsumed under the label "CBT" in the results and discussion. Given that the aim is to discern which treatments are effective for people living with CF, we must distinguish the related treatments accordingly.
For example, p6, line 219 - O'Hayer et al. and Bathgate et al. are both attributed to telehealth-delivered CALM (Bathgate's study) - this needs to be corrected to include that O'Hayer et al.'s study is telehealth-delivered ACT.
Also, O'Hayer et al. is listed in the table as "no gender specified" - gender is in fact specified in the RCT publication.
Fig 2: O'Hayer et al. is attributed to the wrong date (should be 2024). The others are correct.
Then in the discussion, ACT seems to be lumped in under "CBT" - this needs to be corrected to enhance clarity for readers. P9, line 289 "..particularly CBT" - this needs to be changed to reflect that ACT was also deemed effective by the author's criteria.
P. 9 last paragraph - all mention is of CBT - ACT needs to be reflected here too, given that it was deemed effective by the authors' criteria.
P 10, line 317: 3 -month follow-up among "CBT interventions" - ACT needs to be specified here as having a 3 month follow-up.
P 10, line 330: "especially CBT" -ACT also needs a mention, e.g. "especially CBT and ACT"
P 10 line 343: ACT needs to be mentioned here as an effective treatment - currently it's not mentioned in that paragraph.
These minor edits will enhance readability and clarity - best reflecting the results of this thoughtful and well-executed study.
Author Response
Thank you for valuing our review and your involvement in improving the quality of the study.
- We have corrected the oversight regarding references 11, 12, and 13, which were unintentionally missed during the manuscript editing process. These references have been updated and replaced with more appropriate sources addressing ACT within their study, highlighted in yellow in the introduction.
- We have revised the Results section to reflect the distinction between the two studies accurately. Specifically, we corrected the attribution of the intervention in O’Hayer et al.'s study, which was mistakenly grouped with Bathgate et al. under the CALM program. The revised text correctly indicates that O’Hayer et al. investigated ACT delivered via telemedicine, while Bathgate et al. studied the CALM program. This correction has been highlighted in yellow in the Results section.
- We have added information regarding participants from a study by O’Hayer et al. 2024 This update has been incorporated into Table 1 and is highlighted in yellow.
- Concerning Figure 2, we have corrected the publication year of the study by O’Hayer et al. The updated figure now accurately reflects the correct year.
- We have revised the text in the Discussion section to improve clarity regarding the classification of therapeutic approaches. Specifically, we corrected the statement to reflect that CBT and ACT were considered adequate. Also, we mention the importance of the ACT in the conclusion. This clarification has been highlighted in yellow in the revised manuscript.
- We revised the discussion and conclusion sections about the follow-up regarding the ACT intervention, and mentioned acceptance therapy as an effective treatment in patients with CF, highlighted in yellow.
Reviewer 2 Report
Comments and Suggestions for Authors
This systematic review evaluated the effectiveness of psychological interventions in enhancing the quality of life for patients with cystic fibrosis. Although the manuscript is generally well-structured, several modifications are required.
- The number of references is inadequate, and some are outdated. Please replace them with more current sources. Increase the reference number to at least 60.
- This systematic review should have a PROSPERO registration number.
- In the abstract, you should present the full term and its abbreviation in parentheses. For example, fibrosis-specific cognitive-behavioral therapy (CF-CBT), Coping and Learning to Manage Stress with CF (CALM). Check these two abbreviations in the main text as well and follow the same format.
- The rationale for the study should be explained more clearly. Please include references to available data and emphasize the existing gaps in the literature.
- In Table 1, please add another column and specify the country in which each study was conducted.
- Extensive proofreading is essential, as some sentences are excessively long.
- It is recommended that instances of "we" or "our" be replaced with phrases such as “current study", "this study" or "present study".
- The introduction is long and should be more concise.
- The discussion section should be expanded to include more references for comparing the results with previous studies.
- At the beginning of the discussion, you should summarize the key findings of the review.
additional comments:
The topic is original and relevant to the field. It addresses a specific gap in the field. There have been no comprehensive and concise systematic reviews in this area. A previous review has been identified (PMID: 39407865), but it was not a systematic review. The conclusions are consistent with the evidence and arguments presented, and they effectively address the main question posed. However, the number of references is insufficient. The table and figures are appropriate.
Author Response
We sincerely appreciate your thoughtful reassessment of our manuscript and your valuable contributions to enhancing the quality of our review.
We have thoroughly revised the manuscript to address all the points raised.
The PROSPERO registration number has been highlighted in pink in the methods section to strengthen the methodological transparency of our systematic review.
In the abstract, we now provide both the full terms and their corresponding abbreviations—for example, fibrosis-specific cognitive-behavioral therapy (CF-CBT) and Coping and Learning to Manage Stress with CF (CALM)—and this format has also been applied consistently throughout the main text.
We have clarified the rationale for the study by referencing up-to-date data and emphasizing the current gaps in the literature that our review seeks to address.
Table 1 indicates the country where each study was conducted, in the first column, providing further contextual relevance.
The manuscript has undergone comprehensive proofreading to improve readability and eliminate overly long sentences.
Additionally, expressions such as “we” or “our” have been replaced with neutral, academic alternatives to enhance the narrative's objectivity.
We have included a clear summary of the review's key findings at the beginning of the discussion to guide the reader.
The introduction has been condensed for clarity and focus. The discussion section has been expanded, incorporating additional references to facilitate a more robust comparison of our findings with those of previous studies.
The number of references has been significantly increased to over 60, and outdated sources have been replaced with more recent and relevant literature.
The revised manuscript highlights all modifications made in response to the reviewer’s suggestions in pink for ease of identification.
Reviewer 3 Report
Comments and Suggestions for Authors
Dear Authors,
Thank you for providing me with the opportunity to read this interesting piece of paper. Below, I have listed my comments:
1) You do a good job listing various psychological interventions—CBT, ACT, MRT—but the content is mostly descriptive. The introduction would benefit from more critical engagement, such as are certain approaches more effective for specific subgroups? or What challenges exist in implementing these interventions in clinical care? How existing literature is fragmented or inconsistent?
2) In the methods, I am not sure why ChatGPT and Grammarly need to be mentioned.
3) Line 149, what counts as "meaningful insights" in qualitative studies? this is vague and subjective.
4) Which keywords did you use to identify the papers? It is important to mention them.
5) Line 167, rather than saying 'We extract information', it is best to say 'Data were extracted…'
6) Did you use any formal tool to assess risk of bias?
7) The flow diagram needs to be reviewed. Why were 136 papers excluded? It is not possible to have 0 in the last tables.
8) In the findings section, it would have been nice to delve a little more in some studies as it is not clear what studies have done, what they have done, and their limitations.
9) In the discussion, you mention that Tai Chi hasn't demonstrated significant results, which is fine, but it’s compared only briefly to mind-body research. Add 1–2 sentences about what the broader literature says about Tai Chi or mindfulness in chronic illness, and how CF-specific studies could build on that.
10) A general comment, when introducing an acronym, you do not need to state the whole phrase again. Just keep using the acronym in the rest of the paper.
I hope this feedback is helpful.
Author Response
We appreciate your recognition of our review and your valuable contributions toward enhancing the quality of our study.
- In the Introduction section, we acknowledge the importance of adopting a more critical approach. However, we could not identify studies demonstrating the superior or inferior effects of psychological interventions on specific subgroups of patients with CF. In response to a previous reviewer’s suggestion, we have expanded the Discussion section to elaborate on the challenges of implementing psychotherapeutic interventions in clinical care. Additionally, we revised the rationale for conducting this review to emphasize that the existing literature integrating psychotherapeutic interventions in treating patients with chronic conditions such as CF is minimal, fragmented, and inconsistent.
- Following the publication guidelines of the Healthcare journal and the editorial policies of MDPI, it is mandatory to disclose the use of any AI tools, if such tools were employed in the development of the manuscript.
- We have revised the sentence referring to "meaningful insights" to provide greater clarity regarding its intended meaning. The updated phrasing now clearly explains the concept used in our context and has been highlighted in blue in the revised manuscript for easy identification.
- At the beginning of the results section, we completed and mentioned the sequence used for searching articles in databases, highlighted in blue.
- In line 167, we corrected the syntagm “we extract…” with “Extracted information from the selected articles.”
- Unfortunately, no specific tool was used to assess the risk of bias in this review. We acknowledge this as a limitation and clearly state it in the manuscript.
- In the paragraph above Figure 1 (highlighted in blue), we clearly explain why we excluded the 136 studies, which have been revised accordingly in the updated manuscript.
- It is true that a more in-depth discussion of the methodologies used in the most relevant included studies would be valuable. However, we focused primarily on the outcomes of psychological interventions and their long-term effectiveness in improving quality of life. We aimed to emphasize these aspects to establish clear directions for future approaches to addressing the psychological needs of patients with cystic fibrosis.
- We have expanded the Discussion section to include supporting evidence on the psychological benefits of Tai Chi for individuals with chronic illnesses. This addition has been highlighted in blue in the revised manuscript.
- We have acknowledged this oversight and corrected it in the revised manuscript.
Round 2
Reviewer 2 Report
Comments and Suggestions for Authors
The authors tried to address the major comments. The manuscript has been revised and shows potential for publication.
Comments on the Quality of English LanguageMinor proofreading is required.
Reviewer 3 Report
Comments and Suggestions for Authors
Thank you for revising the manuscript.